# Synergic Effect of Microorganism and Colloidal Biochar-Based Organic Fertilizer on the Growth and Fruit Quality of Tomato

**Shiguo Gu [1], Fei Lian [2,*], Hanyue Yang [1], Yaru Han [1], Wei Zhang [1], Fan Yang [1] and Jie Gao [3]**

1    Institute of Environmental Processes and Pollution Control, School of Environmental and Civil Engineering, Jiangnan University, Wuxi 214122, China; ychgushiguo@163.com (S.G.); yang18861831936@gmail.com (H.Y.); xxhanyaru@163.com (Y.H.); zw19952730051@163.com (W.Z.); fanyang1126@163.com (F.Y.)
2    School of Energy and Environmental Engineering, Hebei University of Technology, Tianjin 300401, China
3    Inner Mongolia Sunture Environmental Technology, Co., Hohhot 010020, China; XC4000400065@163.com
*    Correspondence: lianfei2000@126.com; Tel.: +86-022-60435775

**Abstract:** It is well known that carbon-based organic fertilizer can effectively promote crop growth and improve nutrient utilization efficiency. However, little is known about the effect of microorganisms on the nutrient availability of carbon-based organic fertilizer. To elucidate the contribution of microorganisms to the agricultural benefit of colloidal biochar-based fertilizer, a 5-month pot experiment was conducted to study the effect of different combinations of Methyltrophic bacillus, colloidal biochar, and organic fertilizer on physical–chemical properties of soil, plant growth, physiological-biochemical reactions, yield, and quality of tomato. The results show that the addition of Methyltrophic bacillus effectively promoted the availability of soil nutrients (such as nitrate nitrogen and available potassium) and increased soil cation exchange capacity; meanwhile, it significantly increased the content of chlorophyll-a (9.42–27.41%) and promoted the net photosynthetic rate (10.86–13.73%) and biomass of tomato fruit (17.84–26.33%). The contents of lycopene, vitamin C, total sugar, and soluble sugar in the fruits treated by the ternary combination of Methyltrophic bacillus, colloidal biochar, and organic fertilizer increased by 58.40%, 46.53%, 29.45%, and 26.65%, respectively. The above results demonstrate that the addition of beneficial microorganisms could further improve the performance of biochar-based fertilizer on plant growth, yield, and fruit quality of tomato. This information provides evidence for the promising performance of microorganism-supported biochar organic fertilizer in agricultural applications.

**Keywords:** microorganism; colloidal biochar; organic fertilizer; tomato; fruit quality

## 1. Introduction

Biochar (BC), a man-made form of black carbon, is produced by pyrolysis or gasification of organic matter without or with limited oxygen and is rich in organic carbon, mineral elements, and inorganic carbonates [1]. BC mainly consists of aromatic hydrocarbons and alkyl structures, which are highly inert and difficult to be oxidized or decomposed by microorganisms and is extremely stable in soil and can persist for hundreds or even thousands of years [2,3]. BC usually has a large number of surface functional groups and a porous structure, which can improve the pH and ion exchange capacity of acidic soil [4,5] and exhibit high capacity of binding and retaining soil nutrients, so as to reduce the loss of nutrients and promote the uptake of nutrients by plants [6–8]. Therefore, a synergic effect is likely to be achieved by loading organic fertilizer into the pores of BC (as a carrier) to prepare BC-based compound fertilizers, which could increase the nutrient content of BC and more effectively regulate the release of nutrient elements into the soil.

Carbon-based compound fertilizer mainly includes three typical types: carbon-based organic fertilizer, carbon-based inorganic fertilizer, and carbon-based organic–inorganic compound fertilizer. Environmentally friendly carbon-based compound fertilizer is capable of saving energy and improving nutrient utilization efficiency, which is an important

development direction for improving the quality and efficiency of chemical fertilizers. Glaser et al. showed that adding BC to compost in a field experiment increased corn yield by 26% compared to pure compost [9]. Combined addition of BC and inorganic fertilizer significantly improved the maize grain yield and biomass yield compared to the unmanned plot in all irrigation treatments [10]. It is thus clear that the combination of organic fertilizer and BC has evident advantages in crop growth and soil improvement than the individual application [11–13]. In recent years, microbial agents have been widely used to improve the crop growth and can also improve soil properties. Methylotrophic bacteria represent an important group of multi-functional plant growth-promoting bacteria. Methylotrophic bacteria utilize the plant waste to product methanol as the source of carbon and energy and enhances plant growth by producing growth hormones, such as indole-3-acetic acid [14] and cytokinins [15]. Simultaneously, it can produce beneficial metabolites and promote plant growth through antagonism, competition, and induction [16,17]. Pot experiments indicated that the plant height and dry weight of tomato treated with T. afroharzianum TM2-4 microbial agent were increased by 36.1% and 32.3%, respectively, compared with the control [18]. Moreover, some reports showed that microbial-organic fertilizers could replace 23–52% of N fertilizer without any loss of yield [19]. However, there are few studies investigating the synergic effects of microorganisms and BC-based organic fertilizer on crop growth and quality. Therefore, we put forward a scientific hypothesis that microbial agents can further accelerate the nutrient transformation in BC-based organic fertilizer, effectively promote the absorption and utilization of nutrients by plants, and improve the fertilizer utilization efficiency.

Therefore, the purpose of this study was to examine the synergic effect (if any) of microorganisms and BC organic fertilizer on the growth and development of plants. Previous studies found that colloidal BC possesses more abundant oxyl groups, higher surface area, and more negative zeta potential than bulk-BC due to its plenty of submicron and even nano-sized BC fractions [20]. Thus, colloidal BC derived from sawdust was selected to prepare the BC-based fertilizer in the present study. Methyltrophic bacillus and tomato were used as typical plant growth-promoting bacteria and plant, respectively. The growth, physiological and biochemical reactions, and fruit quality of tomato were determined based on a 5-month pot experiment. The results can not only provide new information for the development of promising BC-based organic fertilizer but also provide theoretical support for the agricultural sustainable development.

## 2. Materials and Experimental Design

### 2.1. Materials

The air-dried tree branches were used as the raw material to prepare BC. After ground and sieved (20 mesh), the wood powder was fed into a lab-scale tubular reactor within a muffle furnace (Tianjin Zhonghuan Electric Furnace Co., Ltd., Tianjin, China) and slowly pyrolyzed (10 $^\circ$C/min) in a $N_2$ atmosphere at 550 $^\circ$C for 120 min. The obtained BC was ground and sieved through a 100-mesh sieve. Colloidal BC was obtained by wet sieving technique [20–22]. Briefly, a certain amount of the BC powder was added to 1 L of deionized water and stirred (150 rpm) for 24 h. After sonication (1 h at 120 W), the suspension was slowly poured through a 1-micron sieve and the remaining particles were carefully washed with deionized water to ensure that most of the desired fractions (<1-micron) can pass through the sieve and the BC in the filtrate was obtained after it was centrifuged (referred to as colloidal BC).

The organic fertilizer was purchased from a local environmental technology company, which was mainly composed of starter, rice bran, and pig manure. Methyltrophic bacillus was obtained from China microbial strain collection center (Serial number, CCTCC-AB2014337). A 0.1 mL volume of raw bacillus was spread on Luria Bertani (LB) liquid medium (4.9 mL), which contained (g/L) tryptone, 10; yeast extract, 5; NaCl, 10; and agar, 18, and the solutions were then incubated in a constant temperature shaking incubator (IGS100, Thermo, Waltham, MA, USA) at 180 rpm and 30 $^\circ$C for 48 h. The colony was

extracted from LB liquid medium and dissolved in sterilized phosphate-buffered saline (PBS, pH 7.4) buffer. The concentration of bacterial solution ($1.0 \times 10^6$ CFU/mL) was determined by measuring the light absorption value at OD600 with a spectrophotometer (Shimadzu, Kyoto, Japan). Tomato seeds, "Jinpeng No. 1", were purchased from Xi'an Jinpeng seeds Co., Ltd. (Xi'an, China) The used soil is brown soil and its basic properties are as follows: pH = $6.34 \pm 0.16$, cation exchange capacity (CEC) $14.12 \pm 1.16$ cmol/kg, organic matter content $35.76 \pm 2.04$ g/kg, nitrate nitrogen (N) $12.74 \pm 0.36$ mg/kg, available phosphorus (P) $12.64 \pm 2.37$ mg/kg, and available potassium (K) $126.5 \pm 9.08$ mg/kg. Basic properties of the colloidal BC and organic fertilizer are shown in Table 1.

**Table 1.** Basic properties of colloidal BC and organic fertilizer.

| Category | CEC (cmoL/kg) | Available K (g/kg) | Available P (g/kg) | Nitrate N (mg/kg) | Organic Matter Content (g/kg) | pH |
|---|---|---|---|---|---|---|
| Colloidal BC | $65.63 \pm 4.08$ | $31.56 \pm 1.63$ | $83.47 \pm 16.02$ | $2.26 \pm 0.08$ | $395.33 \pm 27.19$ | $8.98 \pm 0.90$ |
| Organic fertilizer | $73.33 \pm 2.45$ | $112.54 \pm 2.73$ | $185.70 \pm 8.17$ | $15.60 \pm 0.86$ | $380.82 \pm 23.03$ | $9.98 \pm 0.87$ |

### 2.2. Experimental Design

The tomato seeds were sorted and disinfected by washing with 5% (*v/v*) NaClO for 10 min and then germinated in a deionized water irrigated vermiculite medium. The seedlings were incubated in a growth chamber, with 12 h photoperiod, 25/20 °C day/night temperature, and 60% relative humidity. Seedlings with consistent growth (the third leaflet) were transplanted into a pot containing 3 kg of dry soil.

A pot experiment was conducted with 6 treatments, including soil control (CK), soil amended by colloidal BC (3% of soil weight) (T1), soil amended by organic fertilizer (3% of soil weight) (T2), soil amended by colloidal BC (1.5% of soil weight) and organic fertilizer (1.5% of soil weight) (T3), soil amended by colloidal BC (3% of soil weight) and microbial bacterial solution (30 mL, $1.0 \times 10^6$ CFU/mL) (T4), soil amended by organic fertilizer (3% of soil weight) and microbial bacterial solution (30 mL, $1.0 \times 10^6$ CFU/mL) (T5), soil amended by colloidal BC (1.5% of soil weight), organic fertilizer (1.5% of soil weight), and microbial bacterial solution (30 mL, $1.0 \times 10^6$ CFU/mL) (T6). For the treatments containing both colloidal BC and organic fertilizer, the colloidal BC was thoroughly mixed with organic fertilizer, composted, and then fermented at a constant temperature of 30 °C for 4 weeks before use. Colloidal BC-based organic fertilizer was modified by adding Methyltrophic bacillus. Briefly, after fermenting, Methyltrophic bacillus was added to the pots with microbial bacterial treatments. The bacterial treatment group was homogenized with 3 kg of the soil at 30 rpm for 30 min.

### 2.3. Characterization of Soil, BC, and Organic Fertilizer

Before blending with soil, the properties of soil, colloidal BC, and organic fertilizer were measured, respectively. CEC was analyzed by sodium acetate extraction flame photometer method [23]. Available K and P were tested by modified kelowna methods [24]. Nitrate N was estimated by ultraviolet spectrophotometry [25]. Organic matter was measured by TOC analyzer [26]. The pH value was determined by potentiometric method and the water–soil ratio (W:V) was 2.5:1.

### 2.4. Measurements of Plant Growth and Fruit Quality

The heights of the tomato seedlings (from the base to the growth point) were measured using a ruler (centimeters). Tomato fruit was harvested after ripening and the yield was calculated. The photosynthetic indexes were measured in the morning (9:00–11:00) and the second leaf was selected from the top of each plant. Each chosen leaf was measured three times and the average of the three readouts was recorded. Various photosynthetic parameters, including net photosynthetic rate (Pn), transpiration rate (Tr), stomatal conduc-

tance (Gs), and intercellular $CO_2$ concentration (Ci), were measured by the photosynthetic instrument (GFS-3000 WALZ) [27].

For chlorophyll content, the middle and upper leaves of tomato were selected and quickly ground into powder with liquid nitrogen, weighed 0.5 g powder into 10 mL centrifuge tube, and then added cold acetone. The homogenate was extracted in the dark at 4 °C for 24 h and was centrifuged at 2500 rpm for 10 min. The supernatant was separated and the absorbances were read at 662 nm (Chlorophyll a) and 646 nm (Chlorophyll b) on Shimadzu UV-1800 spectrophotometer (Shimadzu, Kyoto, Japan) [28]. Tomato fruit quality was determined by selecting mature fruits (repeat picking 4 plants) with the same growth status of the first ear and the third ear and the average value was taken as the final value. The total sugar in tomato was obtained by anthrone colorimetry [29]. The soluble sugar was determined by sulfuric acid anthrone colorimetry [30]. Soluble protein content was measured using Coomassie Brilliant Blue G-250 (Shanghai Macklin Biochemical Co., Ltd., Shanghai, China) according to the method of Bradford [31]. Vitamin C content was determined using the method of Deutsch and Weeks [32]. Lycopene content was obtained from spectrophotometry at 502 nm [33]. The concentration of soluble solids in tomato was measured by means of a digital refractometer PR-32a (ATAGO Co. Ltd., Fukui, Japan) [34].

### 2.5. Statistical Analysis

Statistical analysis of the growth curve data was performed using standard analysis of variance (ANOVA), followed by a one-way ANOVA with Duncan's Multiple Range Test (DMRT) and were analyzed using SPSS version 21.0 (IBM, Armonk, NY, USA). All the experiments were carried out at least in triplicate and a significant difference was considered at $p < 0.05$.

## 3. Results and Discussion

### 3.1. Effect of Colloidal BC-Based Organic Fertilizer on Physical and Chemical Properties of Soil

The CEC of T4 and T5 treatments was significantly higher than that of T1, which may be enhanced by the self-metabolism of microorganisms in the processes of both compost fermentation and plant growth. Note that the treatment of T6 (i.e., 1.5% colloidal BC + 1.5% organic fertilizer + Methyltrophic bacillus) exhibited the highest CEC values among the different treatments. With the increasing incubation time of colloidal BC in soil, more oxygen-containing functional groups can be formed on its surfaces under the modifications of biotic and abiotic (chemical oxidation) processes, which increased the surface charge density and thereby effectively improved the soil CEC [35,36]. Organic fertilizer had a large number of humus and oxygen-containing functional groups and its surfaces might have a variety of binding sites for cations, which also improved the CEC of soil [37]. Figure 1b shows that the addition of BC or organic fertilizer (T1–T3) significantly ($p < 0.05$) enhanced the content of nitrate N compared to CK. However, the presence of Methyltrophic bacillus (T4, T5, and T6) did not significantly increase the content of nitrate N, relative to treatments without Methyltrophic bacillus, which may be due to the fact that it would take a longer time for microorganisms to adapt to the soil environment.

The contents of available K and P in all the treatments were increased, where the values in the T6 treatment were the highest (Figure 1c,d) compared with CK. The microorganism inoculated treatments (T4, T5, and T6) had a higher available K content relative to the non-microorganism treatments (T1, T2, and T3), which was probably attributed to the conversion of K from the insoluble species to the soluble species through the metabolism process in microorganisms (Figure 1c). However, the growth of microorganisms needs to consume energy, which may consume some available K. The opposite trend is observed for the soil available P (Figure 1d), which can be explained by the fact that microorganisms cannot significantly increase the content of available P in a short incubation time. The data demonstrated that the lack of the organic fertilizer in soil media significantly decreased available P, which was consistent with previous studies [38]. These results indicated that

the microorganism amendment can greatly improve the contents of CEC, nitrate N, and available K in the soil, especially for CEC and available K.

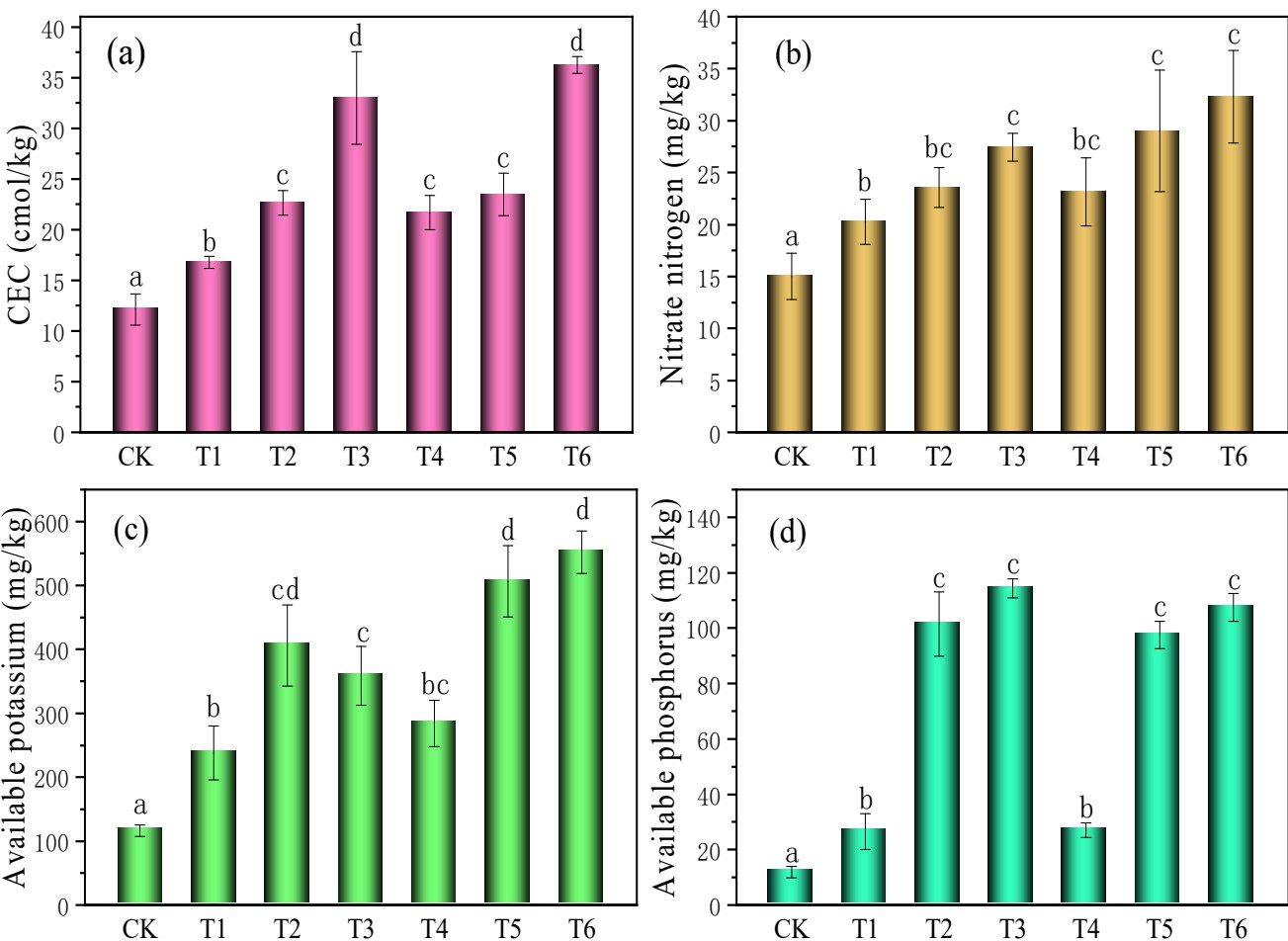

**Figure 1.** Effects of different treatments on soil CEC (**a**), nitrate N (**b**), available K (**c**), and available P (**d**). Soil control (CK), the amount of colloidal BC accounts for 3% of dry soil weight (T1), the amount of organic fertilizer accounts for 3% of dry soil weight (T2), 1.5% colloidal BC +1.5% organic fertilizer (T3), 3% colloidal BC + Methyltrophic bacillus (T4), 3% organic fertilizer + Methyltrophic bacillus (T5), 1.5% colloidal BC + 1.5% organic fertilizer + Methyltrophic bacillus (T6). Data with different letters are significantly different ($p < 0.05$).

### 3.2. Effect of Colloidal BC-Based Organic Fertilizer on the Growth of Plants

As shown in Figure 2a, there was little difference in the growth of plants for each treatment in the first three weeks; however, a significant growth increase occurred from the fourth week. After that, the tomato maintained a rapid growth trend, where the plant height in all the treatments was significantly higher than that in the CK. We found that the growth of tomato reached a peak in the 9th week and then decreased gradually.

For the fruit, the total biomass of tomato fruit in all treatments was significantly higher than that of CK and the fruit biomass in the microorganism inoculated treatments (T4, T5, and T6) was significantly higher than that of non-microorganism inoculated treatments (T1, T2, and T3). It can be seen in Figure 2c that the tomato yield of T6, T5, and T4 treatments were 26.33%, 18.11%, and 17.84% higher than that of T3, T2, and T1 treatments, respectively, indicating that the combined application of microorganism and colloidal BC-based organic fertilizer can significantly promote the growth of plants and improve the yield of tomato simultaneously.

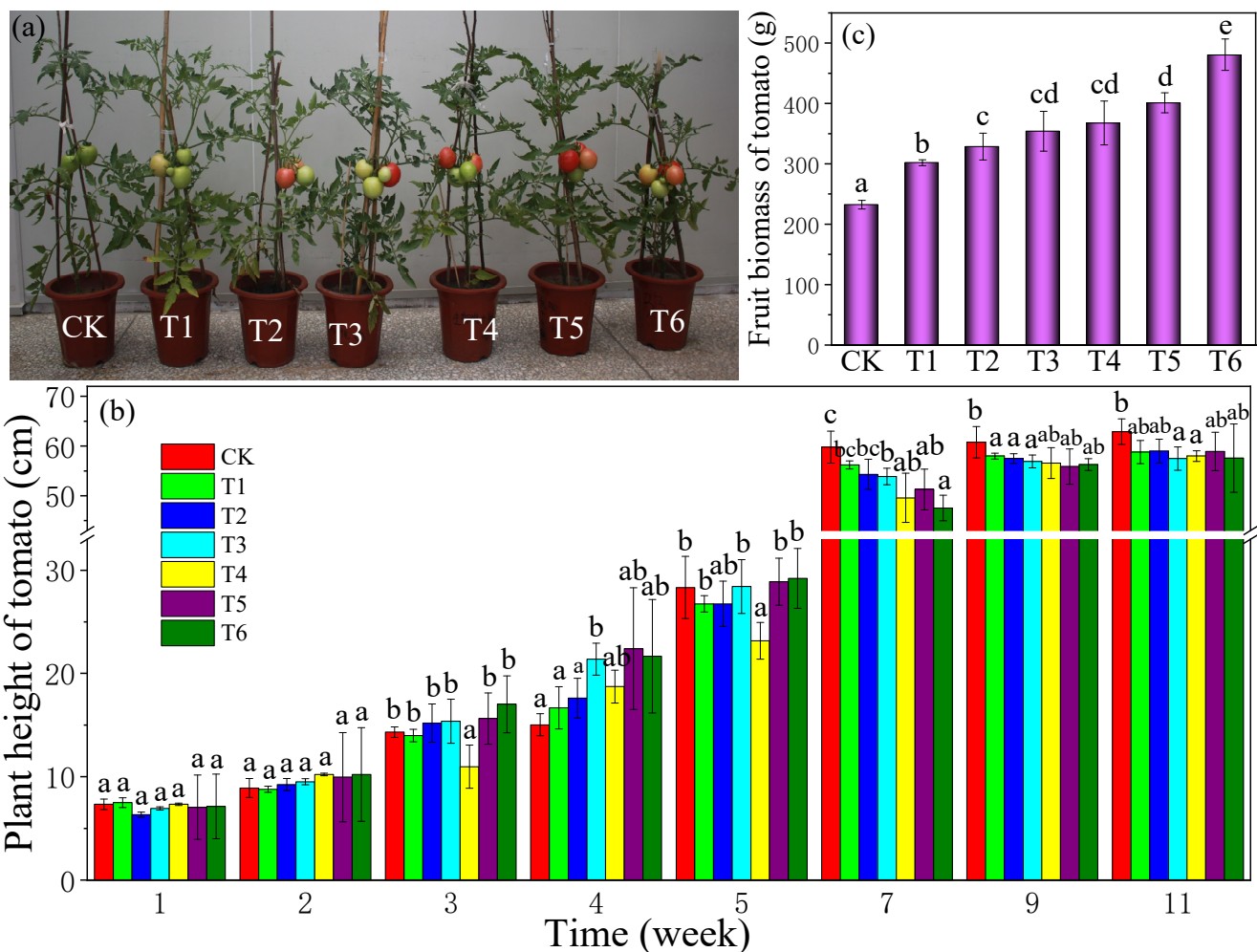

**Figure 2.** Effect of different treatments on growth status (**a**), plant height (**b**), and fruit biomass (**c**) of tomato. Data with different letters are significantly different ($p < 0.05$).

### 3.3. Effect of Colloidal BC-Based Organic Fertilizer on Photosynthesis of Plants

As shown in Figure 3a, the net photosynthetic rate of tomato leaves in all the treatments reached a significant increase (up to 22.01–54.03%) compared with that of CK where the T6 treatment had the largest value. Similarly, the leaf transpiration rate of tomato in all the treatments (except T1) was also significantly higher than that of CK (Figure 3b), among which T6 treatment was the highest (11.69 mmoL $H_2O$ /(m²s)). The leaf transpiration rate of T4 was significantly higher than that of single application of BC ($p < 0.05$) and the T5 treatment increased the leaf transpiration rate of tomato by 10.66%. Compared with only BC-based organic fertilizer treatment, tomato leaf transpiration rate increased by 11.21% under the T6 treatment (i.e., 1.5% colloidal BC + 1.5% organic fertilizer + Methyltrophic bacillus). Moreover, the net photosynthetic rate of tomato leaves was in general, positively correlated with the amount of available P (y = 0.10x + 12.75, $R^2$ = 0.87), which significantly increased tomato photosynthesis. Therefore, available P could also significantly improve the net photosynthetic rate of tomato [39].

The stomatal conductance of tomato leaves in all treatments increased significantly compared with CK (Figure 3c), where T6 was the highest and reached to 461.33 mmol $H_2O$ /(m²s). Compared with the non-microorganism inoculated treatments (T3, T2, and T1), the air pore conductance in the microorganism inoculated treatments (T6, T5, and T4) was significantly increased by 9.18–30.86%. It can be seen from Figure 3b,c that the trend of stomatal conductance and transpiration rate of each treatment was basically consistent. The intercellular $CO_2$ concentration of tomato leaves in each treatment was significantly

lower ($p < 0.05$) than that of CK. Meanwhile, the intercellular $CO_2$ concentration of each treatment decreased by 1.87–11.46% compared with the control. The lower intercellular $CO_2$ concentration indicates that more $CO_2$ might be used in photosynthesis by leaves. Moreover, compared with T3 and T2 (without microorganisms inoculated), the intercellular $CO_2$ concentration of T6 and T5 increased by 5.56% and 4.96%, respectively. There was no significant difference in the intercellular $CO_2$ concentration between T1 and T4 treatments. The change trend of intercellular $CO_2$ content in these treatments was basically opposite to the net photosynthesis rate of leaves, which was in line with the basic rules of plant photosynthesis.

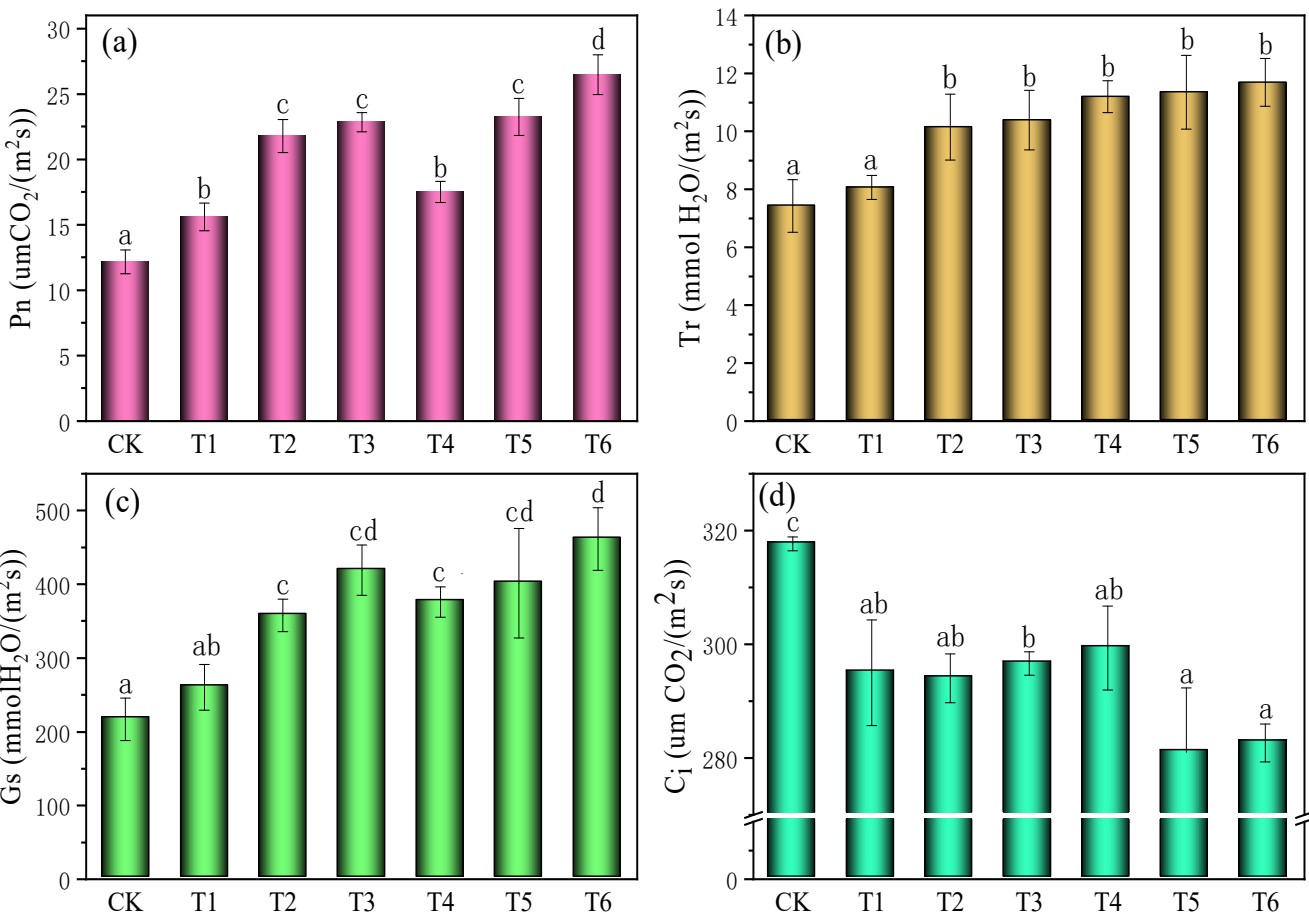

**Figure 3.** The gas exchange parameters: (**a**) net photosynthetic rate (Pn), (**b**) transpiration rate (Tr), (**c**) stomatal conductance (Gs), and (**d**) intercellular $CO_2$ concentration (Ci) of tomato. Data with different letters are significantly different ($p < 0.05$).

According to Figure 3, it is concluded that the application of colloidal BC, organic fertilizer, and microorganisms could actively promote the photosynthesis of tomato leaves. Previous studies indicated that the application of colloidal BC can significantly improve the photosynthesis of seedlings and the application of colloidal BC and organic fertilizer can also significantly improve the transpiration rate of leaves [40,41]. Microorganism inoculation (T4, T5, and T6) had an evident promoting effect on tomato photosynthesis where the photosynthesis of T6 treatment was the highest, indicating that the pores of BC might provide extra living space for the microorganisms and reduce their competition with each other [42]. Comparing the Pn of T4 and T1 with T5 and T2, it was found that the Pn of microbial solution containing colloidal BC is nearly 0.5 $\mu m$ $CO_2/(m^2s)$ higher, which further indicated that colloidal BC (porous) provided growth space for microorganisms. In addition, microorganisms can help the BC-based fertilizer to provide more nutrient elements for plant growth.

### 3.4. Effects of Colloidal BC-Based Organic Fertilizer on Chlorophyll Content of Plants

It is found from Figures 3a and 4 that the content of chlorophyll was highly dependent on the net photosynthetic rate of tomato. The chlorophyll-a contents in the microorganism inoculated treatments (T6, T5, and T4) were 27.41%, 12.74%, and 9.42% higher than those of non-microorganism inoculated treatments (T3, T2, and T1), respectively. Some studies have reported that the application of BC and organic fertilizer can effectively improve the content of chlorophyll in plants and the increase of chlorophyll content can effectively promote photosynthesis [43,44]. In addition, the inoculation of microorganisms also promoted the synthesis of chlorophyll in tomato, where the molecular mechanism still needs to be further studied. Figure 4b shows that the chlorophyll a/b value of tomato was higher than that of the CK, indicating that the microorganisms cooperated with BC-based organic fertilizer to promote the synthesis of chlorophyll in tomato leaves.

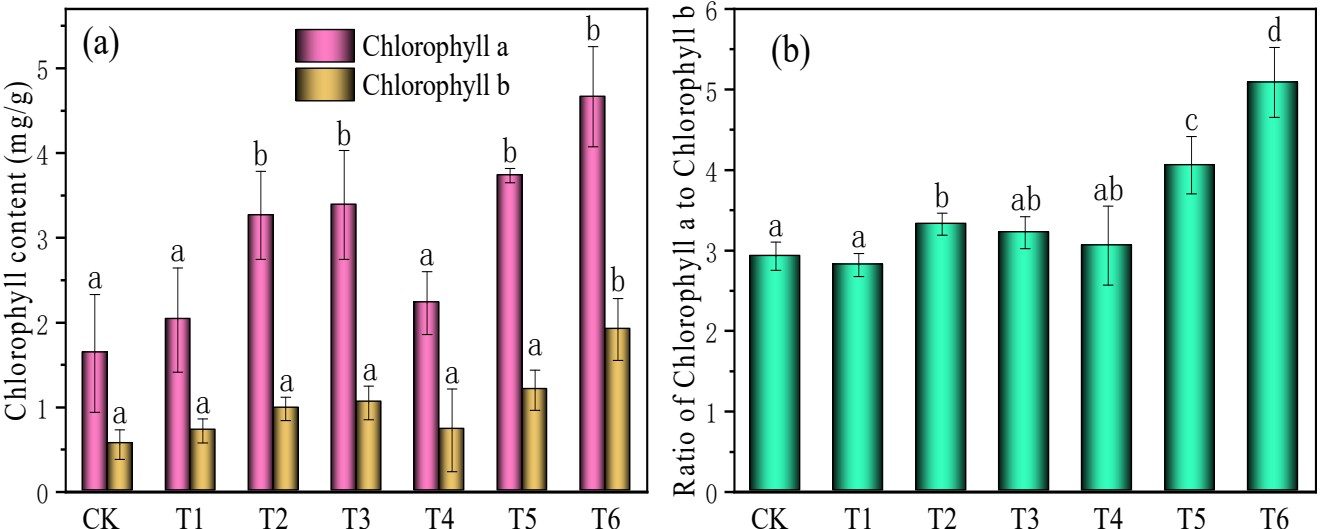

**Figure 4.** Effects of different treatments on chlorophyll content (**a**) and ratio of chlorophyll a to chlorophyll b (**b**) of tomato. Data with different letters are significantly different ($p < 0.05$).

### 3.5. Effects of Colloidal BC-Based Organic Fertilizer on Tomato Quality

Data on the quality of tomato fruits is presented in Table 1. The total sugar content of fruits in the microbial treatments (T5 and T6) was significantly higher ($p < 0.05$) than that of the non-microbial treatments (T2 and T3), for which the total sugar content of tomatoes increased by 39.95% and 30.87%, respectively (Table 1). Table 1 shows that there was no significant difference in soluble sugar of tomato fruit in T2 and T3 treatments. However, for the microorganism inoculated treatments, the soluble sugar content of fruit was 26.65% higher than that in only BC-based fertilizer treatment. It was found that there were no significant changes in soluble protein and soluble sugar contents of tomato in different treatments (Table 1). BC-based fertilizer can provide N for tomato growth and Methyltrophic bacillus can further improve the tomato plants to absorb N or improve the availability of N in BC-based fertilizer for plant growth. It has confirmed that the appropriate increase of N fertilizer can increase the content of soluble sugar in tomato fruits [45,46].

Compared with CK (Table 1), the content of lycopene in different treatments showed an increasing trend and the order from high to low was T6 > T5 > T2 > T3 > CK > T1 > T4. Among them, the lycopene in T6 treatment was 66.39% and 46.56% higher than that of CK and T5, respectively. The vitamin C content of tomato fruit in all treatments was significantly higher than that in the CK ($p < 0.05$). In addition, T6 treatment increased by 21.95% compared with T5, indicating that Methyltrophic bacillus combined with BC-based fertilizer can increase the vitamin C content. It can be seen from Table 2 that the

soluble solid content of tomato fruits in different treatment groups (T2, T3, T4, T5, and T6) showed significant differences compared with CK ($p < 0.05$). Among them, the soluble solid content of fruits treated with T4, T5, and T6 was increased by 8.99%, 12.09%, and 5.81%, respectively, indicating that the synergistic effect of inoculated microorganisms was evident and could greatly increase the soluble solid content of fruits. Maryam et al. also obtained a similar conclusion that organic-based fertilizer can increase the content of soluble solids in tomato fruit [47]. The changes of soil physical and chemical properties under different treatments showed that the CEC and available P of T6 were significantly increased, which was to an extent accounted for in the increased tomato yield, fertilizer utilization efficiency, and fruit quality.

**Table 2.** Effect of different treatments on tomato quality.

| Treatment | Total Sugar (g/kg) | Soluble Sugar (%) | Soluble Protein (g/kg) | Lycopene (mg/100 g) | Vitamin C (%) | Soluble Solids (%) |
|---|---|---|---|---|---|---|
| CK | 28.13 ± 2.01[a] | 5.03 ± 3.21[a] | 12.31 ± 1.71[bc] | 3.25 ± 0.22[ab] | 0.17 ± 0.03[a] | 4.58 ± 0.15[a] |
| T1 | 34.53 ± 0.80[ab] | 7.32 ± 1.21[b] | 10.34 ± 0.38[a] | 3.47 ± 0.28[ab] | 0.34 ± 0.07[b] | 5.18 ± 0.26[b] |
| T2 | 41.37 ± 2.47[b] | 12.03 ± 2.04[bc] | 13.54 ± 0.88[b] | 4.17 ± 0.47[b] | 0.41 ± 0.03[b] | 5.38 ± 0.78[c] |
| T3 | 55.38 ± 1.79[c] | 12.68 ± 0.25[bc] | 13.80 ± 0.51[b] | 4.03 ± 0.14[ab] | 0.32 ± 0.07[b] | 5.67 ± 0.68[bc] |
| T4 | 54.27 ± 1.43[c] | 14.64 ± 0.33[b] | 14.94 ± 2.28[c] | 2.48 ± 0.17[a] | 0.30 ± 0.01[b] | 6.02 ± 1.24[d] |
| T5 | 68.89 ± 0.93[c] | 15.54 ± 0.33[bc] | 14.71 ± 0.76[bc] | 5.17 ± 0.06[c] | 0.47 ± 0.11[b] | 6.12 ± 0.90[c] |
| T6 | 78.51 ± 0.25[d] | 17.29 ± 0.09[c] | 14.72 ± 0.99[c] | 9.68 ± 0.23[c] | 0.61 ± 0.03[c] | 6.23 ± 0.58[d] |

Different letters in the same column indicate significant differences ($p < 0.05$) among treatments.

## 4. Conclusions

Colloidal BC-based organic fertilizer loaded with Methyltrophic bacillus effectively increased the soil CEC value (3.24–24.12 cmoL/kg), nitrate N (3.30–17.28 mg/kg), available K (45.56–435.36 g/kg), and P contents (6.04–95.58 g/kg, except T3), indicating that the addition of growth-promoting bacteria greatly contributed to the transformation and absorption of soil nutrients and then promoted the growth and development of tomato. The combined application of plant growth-promoting bacteria and colloidal BC-based organic fertilizer significantly enhanced the photosynthetic efficiency (Pn, 3.24–14.30 umCO$_2$/(m$^2$s)) of tomato leaves and further improved the quality of tomato fruit (such as total sugar, soluble protein, lycopene, vitamin C, and soluble solids), indicating that the growth-promoting bacteria and colloidal BC-based organic fertilizer had an obvious synergistic effect on the growth of tomato.

From the perspective of soil available N and fruit quality, these results suggest that the combined application of growth-promoting microorganisms and colloidal BC-based organic fertilizer can significantly improve the yield and quality of commercial crops, which has profound practical significance and application values. Meanwhile, it also provides a scientific basis for improving the quality and efficiency of BC-based organic fertilizer and the coupling utilization of microorganisms and BC materials.

**Author Contributions:** Conceptualization, S.G.; Formal analysis, S.G., H.Y. and J.G.; Investigation, W.Z. and F.Y.; Methodology, S.G., H.Y., Y.H., W.Z., F.Y. and J.G.; Project administration, F.L.; Resources, F.L.; Supervision, F.L.; Validation, Y.H.; Writing—review & editing, S.G. and F.L. All authors have read and agreed to the published version of the manuscript.

**Funding:** The work was supported by the National Natural Science Foundation of China (41977278, 41573127), Jiangsu Agricultural Science and Technology Innovation Project (CX (20)3080).

**Institutional Review Board Statement:** Not applicable.

**Informed Consent Statement:** Not applicable.

**Data Availability Statement:** Data sharing is not applicable to this article.

**Conflicts of Interest:** The authors declare no conflict of interest.

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
