# Peer review of "Synergic Effect of Microorganism and Colloidal Biochar-Based Organic Fertilizer on the Growth and Fruit Quality of Tomato"

_coatings, doi:10.3390/coatings11121453_

Round 1

Reviewer 1 Report

It may be published with a series of corrections

Reviewer 2 Report

The paper is well written and will be useful for the researcher in developing biochar-based soil amendments. The manuscript fits the scope of the special issue. However, there is some concern that needs to be cleared.

Line 36. It is unclear what “improve pH” means. Please clarify.

Lines 99-100. A more detailed description of the bacterial preparation is of need.

Lines 110-121. Please describe how the bacterial suspension was introduced into the soil. It is unclear how 30 mL was uniformly distributed within 3 kg of the soil.

Please check the upper and lower indexes throughout the text.

Reviewer 3 Report

84, 119 - 10 °C/min – Please remove the space. Apply this throughout the manuscript.

94 – „g/kg” – The suggested notation is g‧kg-1. Please apply this to all items in the manuscript.

103 – I suggest presenting the properties of the fertilizers used in a tabular form.

128 – „W: V” – Please remove the space.

299 – Please standardize the formatting in the table.

Reviewer 4 Report

Review: Synergic effect of microorganism and colloidal biochar-based organic fertilizer on the growth and fruit quality of tomato.

This manuscript describes the implications of adding Methyltrophic bacillus microorganisms to an organic colloidal biochar-based organic fertilizer on plant nutrient availability and plant growth in general.

The experimental methodology investigates the involvement of the treatments in a wide range of aspects: the fruit biomass, the nutrient availability (nitrate, potassium and phosphorous specifically) and the gas exchange parameters. All these parameters are loyal descriptors of plant development.

Material and experimental design are correctly described and data are well organized. However, I have some mayor comments related to the methodology of the experiments and related to some statements written in the article.

The purpose of the study was to examine the synergic effect of microorganism and BC organic fertilizer on the growth and development of plants. In the experimental design, six treatments are described to assess this hypothesis: a soil control (CK), soil amended by colloidal BC (T1), soil amended by organic fertilizer (T2), soil amended by colloidal BC and organic fertilizer (T3), soil amended by colloidal BC and microbial bacterial solution (T4), soil amended by organic fertilizer and microbial bacterial solution (T5), soil amended by colloidal BC, organic fertilizer and microbial bacterial solution (T6).

In order to mechanistically analyse which is the role of the microbial bacterial solution in plant growth and development, I think you should consider adding the treatment of the soil amended by the microbial bacterial solution. In the following table, there is a logical schema of the combination of the treatments.

Control treatment

Independent treatments

Combination of treatments into pairs

Combination of three treatments

a soil control (CK)

soil amended by colloidal BC (T1)

soil amended by colloidal BC and organic fertilizer (T3)

organic fertilizer and microbial bacterial solution (T6)

Soil amended by organic fertilizer (T2)

soil amended by colloidal BC and microbial bacterial solution (T4)

soil amended by microbial bacterial solution?

soil amended by organic fertilizer and microbial bacterial solution (T5)

In the section 1.1, the effects of colloidal BC based organic fertilizer on physical and chemical properties of soil are described. Although the results are correctly described, I think you could make the effort of explaining other interesting effects that can be observed in the figures. Regarding the influence of the treatments in the availability of phosphorous (figure 1d), data showed that the lack of the organic fertilizer in soil media decreased significantly this availability. If we move to analyse data presented in figure 3a, it seems that there could be a relationship between these parameters.

I recommend these two articles regarding to phosphorous availability in the presence of organic matter.

Erro J., Urrutia O., Baigorri., Aparicio-Tejo P., Irigoyen I., Torino F., Mandado M., Yvin JC, García-Mina JM (2012). Organic complexed superphosphates (CSP): physicochemical characterization and agronomical properties. J. Agric. Food Chem. 2012, 60, 2008-201.

Giovannini C., García-Mina JM., Ciavatta C., Marzadori C. (2013). Effect of organic-complexed supherphosphates on microbial biomass and microbial activity of soil. Biology and Fertility of soils 49, 395-401

In the section 3.3, where you described the effect of colloidal BC based organic fertilizer on photosynthesis of plants, analysing the figure 3a (referred to Pn), I see two effects. The first one is that the treatment responsible of improving Pn is the addition of the organic fertilizer as an amendment. In fact, T2, T3, T5 and T6 treatments showed significant differences compared to control, T1 and T4 treatments. And the second effect, as you explained in the text, is that the combination of the colloidal BC, the organic fertilizer and the microbial bacterial solution enhanced plant Pn significantly compared to all the rest of the treatments.

As for the statement: “the pores of BC might provide extra living space for the microorganisms and reduce their competition with each other”, you will prove this hypothesis comparing the Pn levels between the new treatment of soil amended with microbial solution and the combination of soil amended with BC and the microbial solution.

Regarding the final section of the article, the conclusions are clearly written and they are describing the data you presented.

Round 2

Reviewer 4 Report

Thank you for all the responses and the modifications within the text.

Author Response: thanks for the suggestion. We mainly focus on whether microbial bacterial synergistic colloidal BC-based organic fertilizer can improve the quality and efficiency of tomato. We use microbial bacterial modified colloidal BC, organic fertilizer and colloidal BC-based organic fertilizer as the treatment group. It is not the purpose of the study and has no reference value for modifying soil alone.

Reviewer response: Could you specifically indicate within the text that you are using a microbial bacterial modified colloidal BC?

Line 111: in order to add more information about the characteristics of the compound you are using in the study, you included the table 1, which was very interesting. However, I think there was a mistake, because you indicated in the text  “basic properties of colloidal BC and organic fertilizer are shown in table 1” and then as the title of the table it was written “basic properties of soil, colloidal BC and organic fertilizer”. I only can see two data lines in the table and I am not sure which treatments you are presenting.

Thank you.  

Author Response

Reviewer response: Could you specifically indicate within the text that you are using a microbial bacterial modified colloidal BC?

Response: thanks for the suggestion. In the revised version, we add “Colloidal BC-based organic fertilizer was modified by adding the Methyltrophic bacillus.”

Colloidal BC-based organic fertilizer was modified by adding the Methyltrophic bacillus. (Line 131)

Line 111: in order to add more information about the characteristics of the compound you are using in the study, you included the table 1, which was very interesting. However, I think there was a mistake, because you indicated in the text “basic properties of colloidal BC and organic fertilizer are shown in table 1” and then as the title of the table it was written “basic properties of soil, colloidal BC and organic fertilizer”. I only can see two data lines in the table and I am not sure which treatments you are presenting.

Response: We thank reviewer very much for this careful review. We carefully checked the manuscript and fixed the mistakes. The title of the table was written “basic properties of colloidal BC and organic fertilizer”. (Line 113)

Table 1 Basic properties of colloidal BC and organic fertilizer
